ℨ | **Open Peer Review** | Clinical Microbiology | Research Article

# Diagnostic disruption during a blood culture bottle shortage: clinical and laboratory adaptation strategies

Haripriya Bansal,[1] Aditi Gupta,[1] Kuntal Kumar Sinha,[1] Kumar S. Abhishek,[1] Vidhi Jain,[1] Amit Kumar Rohila,[2] Bharat Choudhary,[2] Deepak Kumar,[3] Vibhor Tak,[1] Mahesh Devnani,[4] Neeraj Gupta,[5] Daisy Khera,[6] Pradeep Kumar Bhatia[7]

**ABSTRACT** In 2024, a global shortage of BD BACTEC blood culture bottles disrupted clinical practices worldwide. This study assessed the impact of the shortage on blood culture performance metrics and evaluated the effectiveness of the management strategies implemented at a tertiary care hospital. This single-center, cross-sectional, observational analysis was conducted at a tertiary healthcare center, spanning from October 2024 to November 2024. During the shortage, clinicians were requested to collect one Microxpress Colorcult culture vial (Tulip Diagnostics), including a 10 mL aerobic culture bottle per patient, with certain exceptions allowed for specific conditions. A total of 4,595 blood cultures were collected during the study period. The overall positivity rate, after excluding contaminated cultures, was found to be 2.8%. *Candida* species and *Brucella* species were isolated less frequently during the shortage period, while no significant changes were observed for other organisms. The adoption of simplified instructions enabled a rapid transition to single blood culture collection during the supply shortage. However, this approach reduced the blood culture positivity rate and hindered the bacteremia rate. These findings highlight the critical need for a stable supply of blood culture bottles and the importance of continued education on optimal blood culture collection practices.

**IMPORTANCE** The 2024 global shortage of BD BACTEC bottles disrupted diagnostic workflows, requiring laboratories and healthcare systems to implement alternative strategies. We evaluated blood culture performance before, during, and after the shortage, and assessed the use of an alternative manual system (Colorcult, MicroXpress, Tulip Diagnostics). While these bottles allowed testing to continue, they showed reduced recovery rates, longer time-to-detection, and higher contamination compared with automated systems. These findings emphasize the diagnostic risks of non-automated alternatives, especially for slow-growing organisms, and underline the need for validated contingency strategies. To our knowledge, this is the first study from India to evaluate such adaptations during a global shortage.

**KEYWORDS** blood culture bottle shortage, Colorcult culture vials, diagnostic yield, blood culture positivity rate

B lood cultures (BC) serve a critical role in diagnosing patients with bloodstream infections (BSI) and associated conditions, including endocarditis, catheter-related bloodstream infections, and sepsis. Blood cultures can identify the microorganisms causing these infections, and follow-on antimicrobial susceptibility testing can help guide optimal therapy (1).

Advances in blood culture systems have increased the yield of blood cultures, reduced the time to organism recovery, and diminished the laboratory technologist's hands-on time. Some automated systems are developed to maximize the recovery

**Peer Reviewer** Matthew A. Pettengill, Thomas Jefferson University, Philadelphia, Pennsylvania, USA

Address correspondence to Vibhor Tak, vibhor_tak@yahoo.com.

The authors declare no conflict of interest.

of fastidious organisms. These systems include the continuous-monitoring "non-invasive" blood culture systems, like the Versa TREK (ThermoScientific, USA), BacT/ALERT (BioMérieux, France), and BACTEC (Becton Dickinson and Company, United States) systems (2).

In June 2024, the manufacturer of BD BACTEC FX Blood Culture System announced a major shortage of blood culture bottles, with a projected supply reduction by 50%, causing delays in diagnosis and managing BSI (3). On 10 July 2024, the Food and Drug Administration (FDA) updated the Medical Device Shortages List to include BC media bottles (4).

Our hospital's microbiology laboratory, which relies on the BD BACTEC Blood Culture System, was directly affected by this shortage. Therefore, this manuscript describes the measures taken by a large academic healthcare system to control blood culture utilization and provide an analysis of the efforts in terms of their impact on utilization and patient care.

## MATERIALS AND METHODS

A cross-sectional, observational analysis was conducted at the All India Institute of Medical Sciences, Jodhpur, Rajasthan, a tertiary healthcare center, spanning from October 2024 to November 2024.

The status of BD BACTEC blood culture bottles was reviewed by our microbiology laboratory committee. Targeted measures were prioritized as only limited number of adult BD BACTEC blood culture bottles were available. Multiple interventions were carried out in tandem. First, considering Colorcult Aerobic culture (Microxpress, Tulip diagnostics) (5) as an alternative, however, these bottles were incompatible with our BD instrument. Next, physician leaders of key hospital departments were notified about the shortage and future blood culture restrictions, thereby encouraging judicious utilization of blood culture bottles. All were encouraged to send only those blood cultures that were suggestive of sepsis. Providers were asked to collect one Colorcult culture bottle, which included a 10 mL aerobic culture bottle. Additionally, a multifaceted training program was implemented for staff to improve blood culture practices. It covered principles of aseptic techniques and proper blood volume inoculation to minimize contamination and maximize pathogen yield. Hands-on phlebotomy simulations and specific protocols for challenging scenarios, such as CVC draws, were also focused. Finally, staff were educated on strict documentation and labeling standards to ensure sample integrity and traceability. The comprehensive training was designed to enhance diagnostic accuracy and improve patient outcomes.

### Laboratory workflow

The received Colorcult bottles were incubated at 35°C–37°C for 5 days for bacteria and 14 days for yeast and fungi or until growth was detected.

Each bottle had a chemical sensor at the bottom that could detect an increase in $CO_2$ produced by the growth of microorganisms. The sensor at the bottom of the bottle was monitored visually for color change, which was proportional to the amount of $CO_2$ present. Growth in the Colorcult (Tulip Diagnostics) culture vials was indicated by the color change of the chemical sensor at the bottom from purple to bright yellow, as shown in Fig. 1. Vials with no growth showed no color change of the chemical sensor and remained purple. The vials were observed for 5 days for a positive color change for bacteria and 14 days for yeast and fungi before reporting them negative (5).

The positive Colorcult (Tulip Diagnostics) culture bottles were subjected to Gram staining and were subcultured on solid media like blood agar, MacConkey agar, and chocolate agar. Identification of organisms was performed using the Vitek MS (bioMérieux) matrix-assisted laser desorption ionization time-of-flight mass spectrometer (MALDI-TOF MS). The negative Colorcult bottles were also subcultured on blood agar after 5 and 14 days to rule out any bacterial or fungal growth, respectively, if present.

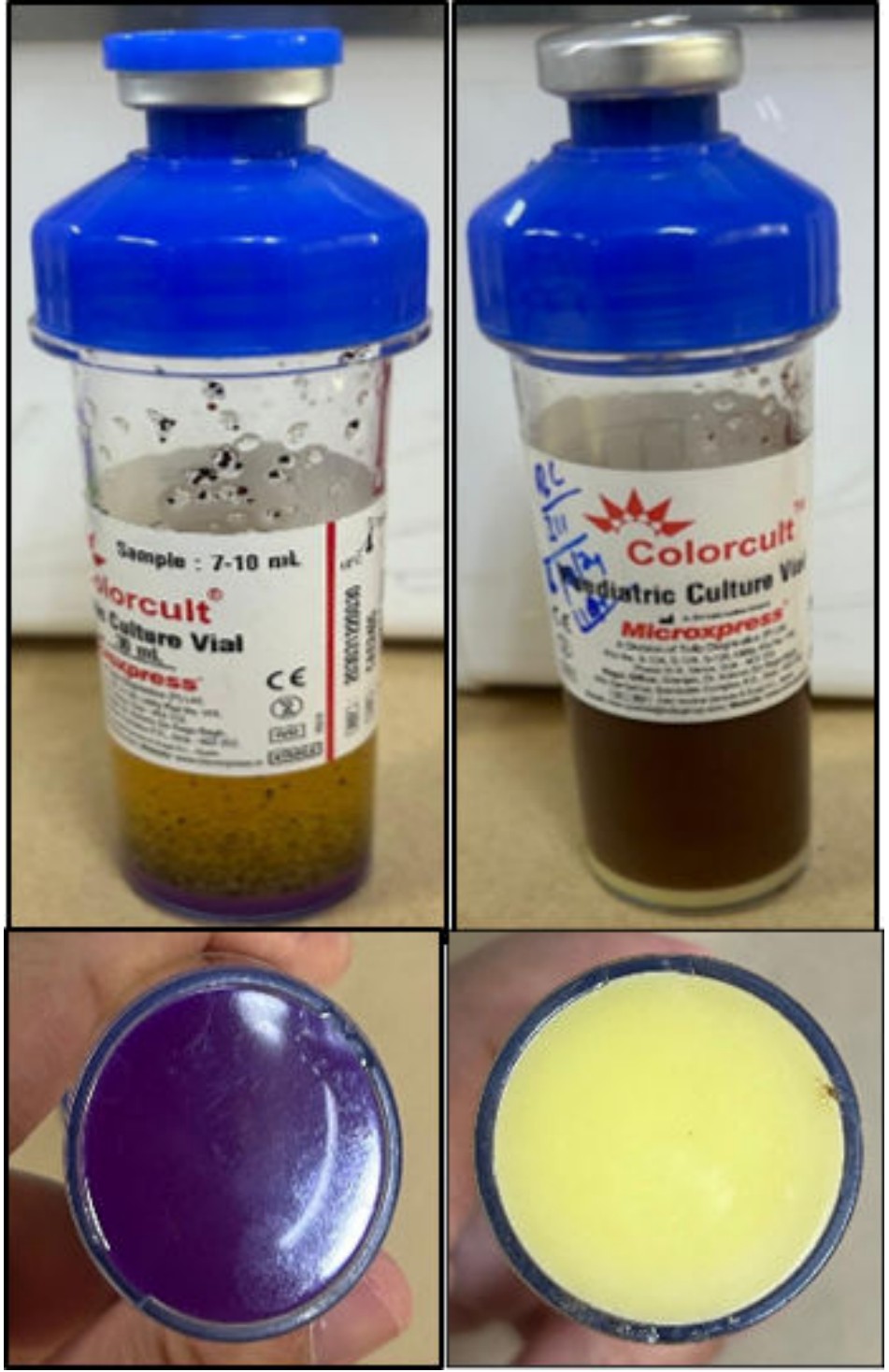

**FIG 1** Color change of the chemical sensor in Colorcult culture vials from purple to bright yellow.

The contamination was defined as any probable contaminant organisms (i.e. *Coagulase-negative staphylococci, Corynebacterium spp., Bacillus spp. [non-anthracis]*, etc.) that were only positive from a single culture. These organisms were considered contaminants unless recovered from multiple blood cultures obtained in sequence, in which case, careful assessment of patients and additional laboratory information was required in defining the significance (6).

In contrast, microorganisms such as *Staphylococcus aureus*, *Streptococcus pneumoniae*, beta-hemolytic streptococci, *Listeria monocytogenes*, *Escherichia coli*, and other members of the *Enterobacterales*, *Pseudomonas aeruginosa*, *Neisseria meningitidis*, *Haemophilus influenzae*, anaerobic gram-negative rods (e.g., *Bacteroides* spp. and *Fusobacterium* spp.), and *Candida* species almost always represent a true infection when isolated from a blood culture (6).

## Statistical analysis and data collection

Since the shortage period lasted for 2 months, data were extracted for the corresponding 2-month intervals immediately before and after the shortage to allow balanced comparison. Positivity and contamination rates across the 3 periods were compared using chi-square tests. Time-to-detection (TTD) was calculated in hours for all positive cultures and expressed as medians with interquartile ranges (IQR). Because TTD distributions were skewed, nonparametric tests were applied. The Kruskal–Wallis test was used for overall comparisons across the 3 periods, and Mann–Whitney U tests were used for pairwise comparisons. A *P*-value < 0.05 was considered statistically significant. All analyses were performed using IBM SPSS Statistics (version 26.0, IBM Corp., Armonk, NY, USA).

The performance of the bottle was evaluated against culture results that were taken as the gold standard. Bottles showing both a positive color change on visual inspection and growth of a pathogenic organism were defined as true positives (TP). Bottles with a color change but had growth of contaminants were considered false positives (FP). Bottles without a color change that later yielded growth on blind subculture were designated false negatives (FN), while bottles without color change and no pathogenic growth were considered true negatives (TN). These classifications were used to calculate diagnostic performance indices, including sensitivity, specificity, positive predictive value (PPV), and negative predictive value (NPV).

## RESULTS

A total of 4,595 blood cultures were received between 1 October 2024 and 30 November 2024 in the Department of Microbiology, All India Institute of Medical Sciences, Jodhpur. This averaged to 75.3 blood cultures per day.

Out of 4,595 bottles, 321 bottles showed color change. The overall positivity rate, after excluding contaminated cultures, from October 2024 to November 2024 was found to be 2.8% (132/4,595). Among the positive cultures (*n* = 132), most common pathogenic organisms found were *Acinetobacter baumannii* (21.2%), *Escherichia coli* (21.2%), and *Pseudomonas aeruginosa* (10.6%), followed by *Klebsiella pneumoniae* (10.6%), *Staphylococcus aureus* (10.6%), *Salmonella* Typhi (6%), *Enterococcus faecium* (4.5%), *Candida tropicalis* (3%), *Burkholderia cepacia* complex (3%), and *Serratia marcescens* (2.2%), and the least common were *Enterococcus faecalis* (1.5%), *Morganella morganii* (1.5%), *Enterobacter hormaechei* (0.7%), *Enterobacter cloacae* (0.7%), *Enterococcus casseliflavus* (0.7%), *Pseudomonas stutzeri* (0.7%), and *Ralstonia mannitolilytica* (0.7%), as shown in Table 1. The contamination rate was found to be 4.1% (189/4,595).

Out of 4,274 Colorcult bottles that did not show any color change, growth on subculture was obtained in 3.2% (139/4,274) samples.

The comparative analysis of diagnostic metrics was conducted across three distinct periods: before the shortage, during the shortage, and after the shortage of blood culture bottles. The key results of this comparison are presented in Table 2.

The overall positivity rate differed significantly across periods ($\chi^2$ = 43.9, *P* < 0.001), decreasing from 9.4% before the shortage (382/4,069) to 7.0% during the shortage (321/4,595) and increasing to 11.1% after (424/3,820). Contamination rates also varied significantly ($\chi^2$ = 14.5, *P* < 0.001), rising from 2.0% before (81/4,069) to 4.1% during the shortage (189/4,595, *P* < 0.001), and subsequently declining to 2.4% after.

Time to detection was most affected. Median TTD was 12 h (IQR 5–20.8) before the shortage, 96 h (IQR 48–96) during the shortage, and 15 h (IQR 11–25) after the shortage.

**TABLE 1** Distribution of pathogenic organisms observed in positive blood culture bottles

| Sr. no | Organism | Number (n) | Percentage (%) |
|---|---|---|---|
| 1 | *Acinetobacter baumannii* | 28 | 21.2% |
| 2 | *Escherichia coli* | 28 | 21.2% |
| 3 | *Pseudomonas aeruginosa* | 14 | 10.6% |
| 4 | *Klebsiella pneumoniae* | 14 | 10.6% |
| 5 | *Staphylococcus aureus* | 14 | 10.6% |
| 6 | *Salmonella Typhi* | 8 | 6% |
| 7 | *Enterococcus faecium* | 6 | 4.5% |
| 8 | *Candida tropicalis* | 4 | 3% |
| 9 | *Burkholderia cepacia* complex | 4 | 3% |
| 10 | *Serratia marcescens* | 3 | 2.2% |
| 11 | *Enterococcus faecalis* | 2 | 1.5% |
| 12 | *Morganella morganii* | 2 | 1.5% |
| 13 | *Enterobacter hormaechei* | 1 | 0.7% |
| 14 | *Enterobacter cloacae* | 1 | 0.7% |
| 15 | *Enterococcus casseliflavus* | 1 | 0.7% |
| 16 | *Pseudomonas stutzeri* | 1 | 0.7% |
| 17 | *Ralstonia mannitolilytica* | 1 | 0.7% |

The Kruskal–Wallis test confirmed a significant difference across the three groups ($\chi^2 = 566$, df = 2, $P < 0.001$). *Post hoc* pairwise comparisons confirmed that TTD during the shortage was significantly prolonged compared with both before and after shortage (all $P < 0.001$). Pairwise Mann–Whitney U tests showed that TTD during the shortage was significantly prolonged compared with both before (U = 5,473, $P < 0.001$) and after (U = 6,730, $P < 0.001$) shortage.

In the statistical analysis of the diagnostic test, TP was 53 bottles and TN was 4,274, while FP and FN were 129 and 139, respectively. PPV was low at 29%, suggesting a positive test was not highly reliable. The NPV was high at 96.85%, indicating the test was very effective at ruling out infection.

Comparing the detection of fastidious and slow-growing organisms, the automated system successfully flagged four *Brucella* isolates within 48–72 h, 1 per month, outside the shortage period. Conversely, none of these isolates were detected using the alternative manual bottles, demonstrating the severe limitation of visual monitoring for such organisms.

## DISCUSSION

Amid a 2-month global shortage of automated blood culture bottles, our tertiary care hospital faced significant diagnostic disruption. We swiftly adapted by revising protocols to collect a single blood culture sample per patient, a critical deviation from standard practice. Concurrently, we implemented robust diagnostic stewardship by educating clinicians and infectious disease specialists on risk stratification. These measures were crucial in managing the crisis, ensuring that limited resources were used judiciously

**TABLE 2** Positivity, contamination, and time-to-detection in different shortage phases

| Outcome | Before shortage (n = 4,069) | During shortage (n = 4,595) | After shortage (n = 3,820) | *P*-value (overall) | Pairwise comparisons *P*-value |
|---|---|---|---|---|---|
| Positivity rate | 382 (9.4%) | 321 (7.0%) | 424 (11.1%) | <0.001 | Before vs during <0.001<br>During vs after <0.001 |
| Contamination rate | 81 (2.0%) | 189 (4.1%) | 90 (2.4%) | <0.001 | Before vs during <0.001<br>During vs after 0.015 |
| Median TTD (hours, IQR) | 12 (5–20.8) | 96 (48–96) | 15 (11–25) | <0.001<br>Kruskal Wallis | Before vs during <0.001<br>During vs after <0.001 |

without compromising patient care. This experience underscores the need for proactive strategies to mitigate the impact of supply chain vulnerabilities on clinical diagnostics.

In our study, during this shortage period, the overall positivity rate was found to be 2.8%, which is lower than those in previous months, where the overall positivity rate, excluding the contamination rate, was 7.4%. Concordant findings were observed in a study done by Naoya Itoh et al. (7), where the observed drop in positivity during the shortage, alongside a simultaneous rise in contamination, suggests a significant under-detection of true pathogens and the missed diagnosis of numerous low-level bacteremia cases. These findings imply that prolonged supply shortages could have a greater impact on patient outcomes, potentially leading to further underdiagnosis of bloodstream infections. After the supply resumed, the positivity rate was found to be 11.1%, which was a significant increase as compared to during the shortage.

The contamination rate significantly increased to 4.1% during the shortage period, compared to 2.0% before and 2.4% after the shortage, which was found to be significantly different. This rise is consistent with literature showing increased contamination when using unfamiliar, manual systems or deviating from standard protocols. Possible reasons include increased manual handling for blind subcultures and rushed procedures due to staff stress and crisis measures. A similar study by Shelly et al. (8) on blood culture stewardship during a shortage supports that changes in protocols necessitated by a crisis can impact quality metrics. The literature supports the fact that any shift from a highly standardized, automated process introduces contamination risk despite adequate training.

The types of microorganisms detected remained largely consistent during the supply shortage period, except for a decrease in Candida species. We did not detect fastidious organisms such as Streptococcus spp. and Brucella species, illustrating the limitation of semiautomated methods and manual interpretation (7). Another study by Kirn et al. (9), which compared a new manual method to an automated system, often highlights the superiority of automation for yield of fastidious organisms.

The most striking effect was on time to detection. Median TTD increased from 12 h (IQR 5–20.8) to 96 h (IQR 48–96) during the shortage, reflecting both the slower detection capacity of the alternative bottles and the limitations of manual visual inspection. Once automated bottles were reinstated, TTD returned to 15 h (IQR 11–25), comparable to baseline.

A study comparing a manual system with the BacT/ALERT 3D automated system found a significant difference in TTD (10). The median TTD was 15.83 h for the automated system, while it was 66.95 h for the manual system. This finding, with a P-value of <0.0001, highlights the significant time delay associated with manual methods, consistent with our report findings (P-value of <0.0001).

The Microxpress Colorcult bottles (Tulip Diagnostics) were not compatible with the BD BACTEC system and required manual interpretation based on color changes, making the readings subjective. Additionally, 3.2% of samples that showed no visible color change grew pathogenic organisms on subculture, representing a major error. The performance of the bottle revealed a low PPV (29%) and high NPV (96.85%), indicating that the bottles were good at ruling out but poor at confirming the presence of a pathogen.

Since such supply restrictions can take place due to various unforeseen reasons, various strategies can be employed in such scenarios. Conventional strategies are considered best practices and should be followed even when supplies are sufficient, like optimizing blood culture volume and set numbers and reducing contamination through proper aseptic sample collection. Contingency strategies are temporary measures that, while not standard practice, are implemented when the benefits outweigh the risks during shortages. These may include reducing blood culture use when other cultures, such as urine for pyelonephritis, bile for cholangitis, or sputum for severe pneumonia, are more effective. Crisis strategies significantly deviate from established practices and are implemented in dire situations, such as using only one blood culture set for the

initial detection of bloodstream infections, utilizing pediatric blood culture bottles, using fungal bottles, or relying solely on anaerobic bottles (11, 12). Additionally, expired blood culture bottles may also be used in such scenarios (13).

We would like to highlight the crucial role of diagnostic stewardship in blood culture practices. Given the severity of sepsis and the presence of multidrug-resistant organisms, blood cultures are often overutilized in diagnosing these infections. A study by Humphries *et al.* (14) assessed the effects of a diagnostic stewardship program and observed a nearly 50% reduction in blood culture rates following its implementation (15). Such programs have a direct impact on antimicrobial stewardship practices, with another study observing a decrease in Days of Therapy (DOT) post-intervention (16). Developing a diagnostic stewardship plan for blood cultures starts with putting together a focused team, reviewing current literature, and closely examining how blood culture tests are ordered within the hospital. Practical steps include setting clear minimum criteria for ordering, limiting paired sample collections, cutting down on unnecessary repeat testing, and focusing on staff education to ensure efficient resource use.

In the middle of a critical supply shortage, our tertiary care center in Western Rajasthan discovered that staff training and diagnostic stewardship cannot overcome the inherent limitations of semi-automated systems. While we implemented alternative collection strategies and prudent utilization measures, the high patient load and incompatibility with our automated instruments led to a demonstrable decline in diagnostic performance. The rapid normalization of metrics upon the reintroduction of automated bottles highlights the critical role of technology and the need for resilient supply chains and contingency plans to safeguard modern diagnostics.

Additionally, despite the limitations, these alternative blood culture bottles represent a practical replacement during supply shortages. Their semi-automated design reduces the reliance on fully manual methods and may also be valuable in low-resource settings where continuous-monitoring blood culture systems are not available.

This study has several limitations. First, the study was conducted at a single tertiary care center over a short 2-month period, which may limit the generalizability of the findings. Second, anaerobic and pediatric bottles were not included in the comparative analysis, which restricts extrapolation. Third, volume data per bottle were not consistently available, and we were not able to assess the adequacy of inoculation. Finally, we were not able to directly assess the impact of this shortage on patient-level morbidity or mortality.

To the best of our knowledge, this is the first report from India to analyze the implementation of diagnostic stewardship and alternative strategies during the BD BACTEC supply shortage. As a tertiary care center in a low- and middle-income country, our institution navigated the crisis with resilience, ensuring uninterrupted reporting and communication of culture results. These efforts facilitated timely treatment decisions and helped prevent complications despite significant diagnostic constraints.

## Conclusion

The crisis affirmed the Colorcult bottles' potential as a resilient, low-resource contingency, effectively preventing a complete diagnostic shutdown during the severe supply shortage. While the limitations of its manual color-change detection led to a TTD and a high FN rate, its swift implementation ensured continuous sample collection. The alternative system served a vital function in a constrained setting, highlighting its utility as an essential, non-automated backup solution. Such bottles can be useful for low-to-middle-income countries and during future supply chain disruptions.

## AUTHOR AFFILIATIONS

[1]Department of Microbiology, All India Institute of Medical Sciences Jodhpur, Jodhpur, India

[2]Department of Trauma and Emergency, All India Institute of Medical Sciences Jodhpur, Jodhpur, India

[3]Department of General Medicine, All India Institute of Medical Sciences Jodhpur, Jodhpur, India

[4]Department of Hospital Administration, All India Institute of Medical Sciences Jodhpur, Jodhpur, India

[5]Department of Neonatology, All India Institute of Medical Sciences Jodhpur, Jodhpur, India

[6]Department of Pediatrics, All India Institute of Medical Sciences Jodhpur, Jodhpur, India

[7]Department of Anaesthesiology and Critical Care, All India Institute of Medical Sciences Jodhpur, Jodhpur, India

## AUTHOR ORCIDs

Haripriya Bansal http://orcid.org/0000-0001-9174-2058
Aditi Gupta http://orcid.org/0009-0005-6728-4751
Vibhor Tak http://orcid.org/0000-0002-1346-7481

## AUTHOR CONTRIBUTIONS

Haripriya Bansal, Conceptualization, Data curation, Writing – original draft | Aditi Gupta, Conceptualization, Data curation, Formal analysis, Writing – original draft, Writing – review and editing | Kuntal Kumar Sinha, Data curation, Methodology | Kumar S. Abhishek, Project administration, Resources | Vidhi Jain, Formal analysis, Validation | Amit Kumar Rohila, Project administration, Supervision | Bharat Choudhary, Project administration, Resources | Deepak Kumar, Supervision | Vibhor Tak, Supervision | Mahesh Devnani, Resources, Supervision | Neeraj Gupta, Project administration, Supervision | Daisy Khera, Supervision, Validation | Pradeep Kumar Bhatia, Supervision, Validation

## ADDITIONAL FILES

The following material is available online.

Open Peer Review

**PEER REVIEW HISTORY (review-history.pdf).** An accounting of the reviewer comments and feedback.

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
