## [Reviewer comments · Microbiology Spectrum]

Microbiology Spectrum

Diagnostic Disruption During a Blood Culture Bottle Shortage: Clinical and Laboratory Adaptation Strategies

Haripriya Bansal, Aditi Gupta, Kuntal Sinha, Kumar Abhishek, Vidhi Jain, Amit Rohila, Bharat Choudhary, Deepak Kumar, VIBHOR TAK, Mahesh Devnani, Neeraj Gupta, Daisy Khera, and Pradeep Bhatia

Corresponding Author(s): VIBHOR TAK, All India Institute of Medical Sciences - Jodhpur

Review Timeline:

Submission Date:	May 29, 2025
Editorial Decision:	July 25, 2025
Revision Received:	September 30, 2025
Accepted:	February 2, 2026

Editor: Arryn Craney

Reviewer(s): Disclosure of reviewer identity is with reference to reviewer comments included in decision letter(s). The following individuals involved in review of your submission have agreed to reveal their identity: Matthew A. Pettengill (Reviewer #2)

Transaction Report:

DOI: <https://doi.org/10.1128/spectrum.01646-25>

Re: Spectrum01646-25 (Diagnostic Disruption During a Blood Culture Bottle Shortage: Clinical and Laboratory Adaptation Strategies)

Dear Dr. VIBHOR TAK:

Thank you for the privilege of reviewing your work. Below you will find my comments, instructions from the Spectrum editorial office, and the reviewer comments.

Revision Guidelines

Sincerely,
Arryn Craney
Editor
Microbiology Spectrum

Reviewer #1 (Comments for the Author):

Thank you for the opportunity to review the report titled "Diagnostic Disruption During a Blood Culture Bottle Shortage: Clinical and Laboratory 1 Adaptation Strategies" which describes the impact of a blood culture bottle shortage on performance metrics in a academic tertiary care center in India. The report indicates an evaluation of the effectiveness of management strategies to address and overcome the shortage was conducted. I found the report to be easy to follow. I have very few comments that

would improve the report.

Overall:

The report concludes that staff education and on blood culture practices is essential for facing the challenges with blood culture shortages however offers very little in terms of data to support this conclusion. Consider softening this language or focusing the conclusion on the results presented for the bottle type including a decrease in recovery rates using the alternative bottle.

It appears that the intervention lasted for 60 days, which means that a pre-post analysis, including the 60 days prior, could provide a clearer understanding of the data including the positivity and contamination rates and any statistical significance between the periods to better evaluate the performance of the alternative bottle during the shortage period.

Introduction:

Line 63-71: Consider moving to the methods, as this paragraph describes the interventions

Line 88-91: Were vials visually read or read by a scanner?

Methods:

Line 110-111: Is there data missing from the Results? This sentence suggests that positivity rates over time, comparing BACTEC to Colorcult, are presented in the results.

Results:

Line 132: Could the authors clarify why 4,314 bottles did not show a color change if 321 out of 4595 had a color change? There appears to be a discrepancy of 40 bottles, so lines 117 and 140 are not aligned.

Consider expanding the table to include those organisms that were recovered from the bottles that did not show a color change. Were there any trends or statistical significance? Did these tend to be the fastidious organisms? If trends were observed consider adding to the discussion as this would be useful to those considering this bottle type as an alternative.

Discussion:

Was the contamination rate consistent with the previous months when BACTEC bottles were used, or did it increase with this bottle type?

While I appreciate the discussion of the importance of diagnostic stewardship, I am unclear how this supports the interventions/results of the report. Was any data collected to demonstrate that the alternative process improved practices or was resource-efficient enough to be shared? The authors could elaborate on what training was provided to the staff in the Methods or the Results if improvements in contamination rates or blood culture volumes were seen.

Limitations were not discussed.

May want to add the following reference:

Shelly MJ, Kunz A, Mittal J, Corwin DS, Chapman C, Ender PT. Efficacy of a Multicenter Hospital Network's Approach to Enacting Blood Culture Stewardship During a Global Shortage. *Open Forum Infect Dis.* 2025 May 16;12(6):ofaf294. doi: 10.1093/ofid/ofaf294. PMID: 40453880; PMCID: PMC12125671.

Reviewer #2 (Comments for the Author):

This is an interesting narrative about how the authors navigated a global supply chain issue related to blood cultures. It needs some additional information to really be a study of the impact. For example almost no information is provided about blood culture data or performance before or after the shortage. The medical staff were instructed to only collect from cases with potential sepsis - in theory this would have increased the % true positives if nothing else changed, selecting higher risk patients, the opposite happened probably due to inferior reagents, lack of anaerobic coverage etc. But it is hard to contextualize without knowing how many blood cultures were collected before the implementation alternate collection advice and alternate media.

Were you able at all evaluate how the collectors were performing with the alternate bottles for mL per bottle? Says 10 mL per bottle was the target, but what happened in practice?

Minor note but fungi indication it says once they were held 14 days, and in another place it says 15 days.

Do you have any data about not only the reduced rate of positivity but the delay in positivity among those that were positive? It seems the color change to the naked eye was difficult to discern at early positivity, so presumably in addition to not detecting some cases at all, the ones that were detected likely were slower than usual for the same organism. If you have time-to-

positivity data available before/during/after the intervention that would be very interesting.

In line 106 Enterobacteriaceae should be changed to Enterobacterales.

In lines 117-141 some of the numbers don't add up. Of 4595 bottles 321 showed color change, of the 321 it is noted that 132 were true positives (line 119) and 149 were contaminants (line 127), and that all bottles with color change grew something (lines 137-138). But $132+149$ does not equal 321, so it seems to me that something is incorrect. Also It is listed that 4314 had no color change (line 133), but $4595-321$ does not equal 4314. It seems the 321 with color change is what is incorrect?

The specificity calculation in line 137 is hard to line up with the other data as well, please show what the calculation is based on.

In lines 152-154 it should be noted that there were a lot of variables. Likely the % reduction in positivity was greater than observed given the complete change in the selection of patients to culture, focusing on high risk.

Brucella is a slow growing organism and sometimes not positive within 5 days even with automated equipment, so it isn't a surprise you didn't detect any with the alternate system. This would be interesting to discuss in further detail. How often did you detect it within 5 days with Bactec?

I think the summary claims in lines 217-220 are not at all supported by what is described in this manuscript. It sounds like a thoughtful process was put in place and was necessary but there is no way to say what the impact was on mortality or morbidity, probably even if you specifically studied that, which you didn't.

REVIEWER RESPONSE

Reviewer Comment	Author Response
Reviewer #1 (Overall): The report concludes that staff education and on blood culture practices is essential for facing the challenges with blood culture shortages however offers very little in terms of data to support this conclusion. Consider softening this language or focusing the conclusion on the results presented for the bottle type including a decrease in recovery rates using the alternative bottle.	We thank the reviewer for this observation. The conclusion has been revised to focus directly on the observed results, particularly the decrease in recovery rates and diagnostic yield with the alternative bottle. The language on staff education has been softened to avoid overstatement. Page 10 – 11, Lines 271 - 277
Reviewer #1 (Overall): It appears that the intervention lasted for 60 days, which means that a pre-post analysis, including the 60 days prior, could provide a clearer understanding of the data, including the positivity and contamination rates and any statistical significance between the periods to better evaluate the performance of the alternative bottle during the shortage period.	We appreciate this suggestion. A comparative analysis has now been added, including two-month periods before, during, and after the shortage. Positivity, contamination, and time-to-detection were compared using chi-square and non-parametric tests. These results are presented in Table 2 and described in the Results section. Pages 7
Reviewer #1 (Introduction): Consider moving to the methods, as this paragraph describes the interventions.	This section has been moved from the Introduction to the Methods. Page 4 – 5, Lines 59 - 74
Reviewer #1 (Introduction): Were vials visually read or read by a scanner?	We have clarified in the Methods that vials were visually inspected for colour change. Page 5, Line 79 - 80
Reviewer #1 (Methods): Is there data missing from the Results? This sentence suggests that positivity rates over time, comparing BACTEC to Colorcult, are presented in the results.	We have clarified the text to ensure that positivity data are clearly presented in the Results, and restructured the sentence for clarity in the Methods. Pages 7 - 8
Reviewer #1 (Results): Could the authors clarify why 4,314 bottles did not show a color change if 321 out of 4595 had a color change? There appears to be a discrepancy of 40 bottles,	We thank the reviewer for identifying this discrepancy. This was due to a typographical error in the previous draft. The corrected values are now consistent

so lines 117 and 140 are not aligned.	across the Results section and tables. Page 6, Line 136
Reviewer #1 (Results): Consider expanding the table to include those organisms that were recovered from the bottles that did not show a color change. Were there any trends or statistical significance? Did these tend to be the fastidious organisms? If trends were observed consider adding to the discussion as this would be useful to those considering this bottle type as an alternative.	We investigated this. Organisms of various types were recovered on blind subculture as well as in bottles with colour change. However, no consistent trend was observed.
Reviewer #1 (Discussion): Was the contamination rate consistent with the previous months when BACTEC bottles were used, or did it increase with this bottle type?	We have clarified in the Results that contamination rates increased significantly during the shortage compared with both before and after, and possible explanations are discussed in the Discussion. Results – Table 2 on Page 7, Line 146 – 150 on Page 8 Discussion – Page 9, Lines 196 - 204
Reviewer #1 (Discussion): While I appreciate the discussion of the importance of diagnostic stewardship, I am unclear how this supports the interventions/results of the report. Was any data collected to demonstrate that the alternative process improved practices or was resource-efficient enough to be shared? The authors could elaborate on what training was provided to the staff in the Methods or the Results if improvements in contamination rates or blood culture volumes were seen.	We revised the Discussion to clarify that staff were trained in aseptic technique, proper volume collection, and documentation during the shortage. However, we also acknowledged that contamination rates still rose, indicating that training alone could not offset the limitations of manual systems. Methods – Page 4-5, Lines 68 – 83 Discussion – Page 9, Lines 203 - 204
Reviewer #1 (Discussion): Limitations were not discussed.	A Limitations subsection has been added on Page 10, Lines 256 – 261.
Reviewer #1 (Discussion): May want to add the following reference: Shelly MJ, Kunz A, Mittal J, Corwin DS, Chapman C, Ender PT. Efficacy of a Multicenter Hospital Network's Approach to	We thank the reviewer for this suggestion. The reference has been added to the revised manuscript.

Enacting Blood Culture Stewardship During a Global Shortage. Open Forum Infect Dis. 2025 May 16;12(6):ofaf294. doi: 10.1093/ofid/ofaf294. PMID: 40453880; PMCID: PMC12125671.	Page 8, Line 191
---	------------------

Reviewer Comment	Author Response
Reviewer #2 (Overall): This is an interesting narrative about how the authors navigated a global supply chain issue related to blood cultures. It needs some additional information to really be a study of the impact. For example, almost no information is provided about blood culture data or performance before or after the shortage. The medical staff were instructed to only collect from cases with potential sepsis - in theory this would have increased the % true positives if nothing else changed, selecting higher risk patients, the opposite happened probably due to inferior reagents, lack of anaerobic coverage etc. But it is hard to contextualize without knowing how many blood cultures were collected before the implementation alternate collection advice and alternate media.	We thank the reviewer for highlighting this gap. Comparative data from the two months before, during, and after the shortage have now been added, including positivity, contamination, and time-to-detection. These are presented in Table 2 and described in the Results section. Page 7
Reviewer #2: Were you able at all evaluate how the collectors were performing with the alternate bottles for mL per bottle? Says 10 mL per bottle was the target, but what happened in practice?	Unfortunately, reliable volume data could not be extracted across all samples. This limitation has been acknowledged in the Discussion. Page 10, Lines 256 – 261
Reviewer #2: Minor note but fungi indication it says once they were held 14 days, and in another place it says 15 days.	Corrected for consistency throughout (14 days). Page 5, Line 90 - 91
Reviewer #2: Do you have any data about not only the reduced rate of positivity but the delay in positivity among those that were positive? It seems the color change to the naked eye was difficult to discern at early positivity, so presumably in addition to not detecting some	We thank the reviewer for this point. Time-to-detection was analysed and added to the Results. Median TTD increased markedly during the shortage compared with both before and after. These findings are included in Table 2 and discussed in the

cases at all, the ones that were detected likely were slower than usual for the same organism. If you have time-to-positivity data available before/during/after the intervention that would be very interesting.	Discussion section. Result: Page 7, Table 2 Discussion: Page 9, Lines 151 - 157
Reviewer #2: In line 106 Enterobacteriaceae should be changed to Enterobacterales.	Corrected. Page 5, Line 99
Reviewer #2: In lines 117–141 some of the numbers don't add up. Of 4595 bottles 321 showed color change, of the 321 it is noted that 132 were true positives (line 119) and 149 were contaminants (line 127), and that all bottles with color change grew something (lines 137–138). But 132+149 does not equal 321, so it seems to me that something is incorrect. Also it is listed that 4314 had no color change (line 133), but 4595–321 does not equal 4314. It seems the 321 with color change is what is incorrect?	We thank the reviewer for catching this discrepancy. The error was due to a typographical error in the earlier draft. The corrected values are now internally consistent and updated in the Results section. Page 6, Line 136
Reviewer #2: The specificity calculation in line 137 is hard to line up with the other data as well, please show what the calculation is based on.	We have clarified definitions of TP, FP, FN, and TN in the Methods. Based on this, PPV was 29% and NPV was 96.85%. These values are now presented clearly in the Results. Methods: Page 6, Lines 113 - 120 Page 8, Line 158 - 161
Reviewer #2: In lines 152–154 it should be noted that there were a lot of variables. Likely the % reduction in positivity was greater than observed given the complete change in the selection of patients to culture, focusing on high risk.	We agree with the reviewer. This has been added to the Discussion as a limitation. Page 8 , Lines 177 - 185
Reviewer #2: Brucella is a slow growing organism and sometimes not positive within 5 days even with automated equipment, so it isn't a surprise you didn't detect any with the alternate system. This would be interesting to discuss in further detail. How often did you	We have added these details: four Brucella isolates were detected by BACTEC within 48–72 hours (outside the shortage). None were detected by the manual bottles. This is now noted in the Results and Discussion. Result: Page 8. Lines 162 – 165

detect it within 5 days with Bactec?	Discussion: Page 9, Line 196 - 201
Reviewer #2: I think the summary claims in lines 217–220 are not at all supported by what is described in this manuscript. It sounds like a thoughtful process was put in place and was necessary but there is no way to say what the impact was on mortality or morbidity, probably even if you specifically studied that, which you didn't.	We appreciate this observation. Statements implying an impact on mortality/morbidity have been removed. The conclusion has been reframed to focus only on observed diagnostic outcomes. Page 10: Line 270 - 276

Re: Spectrum01646-25R1 (Diagnostic Disruption During a Blood Culture Bottle Shortage: Clinical and Laboratory Adaptation Strategies)

Dear Dr. VIBHOR TAK:

Your manuscript has been accepted, and I am forwarding it to the ASM production staff for publication. Your paper will first be checked to make sure all elements meet the technical requirements. ASM staff will contact you if anything needs to be revised before copyediting and production can begin. Otherwise, you will be notified when your proofs are ready to be viewed.

Sincerely,
N. Esther Babady
Editor
Microbiology Spectrum

Reviewer #1 (Comments for the Author):

All reviewer comments have been thoroughly and accurately addressed in the revised manuscript, including corresponding textual, figure, and reference corrections.